# Hybrid 3D Shape Measurement Using the MEMS Scanning Micromirror

**DOI:** 10.3390/mi10010047

**Published:** 2019-01-11

**Authors:** Tao Yang, Guanliang Zhang, Huanhuan Li, Xiang Zhou

**Affiliations:** 1School of Mechanical Engineering, Xi’an Jiaotong University, Xi’an 710049, Shaanxi, China; xjtu.yangtao@gmail.com (T.Y.); gl-zhang@foxmail.com (G.Z.); xjtu.vivianli@gmail.com (H.L.); 2School of Food Equipment Engineering and Science, Xi’an Jiaotong University, Xi’an 710049, Shaanxi, China

**Keywords:** MEMS scanning micromirror, fringe projection, laser stripe scanning, quality map, large reflection variations

## Abstract

A surface with large reflection variations represents one of the biggest challenges for optical 3D shape measurement. In this work, we propose an alternative hybrid 3D shape measurement approach, which combines the high accuracy of fringe projection profilometry (FPP) with the robustness of laser stripe scanning (LSS). To integrate these two technologies into one system, first, we developed a biaxial Microelectromechanical Systems (MEMS) scanning micromirror projection system. In this system, a shaped laser beam serves as a light source. The MEMS micromirror projects the laser beam onto the object surface. Different patterns are produced by controlling the laser source and micromirror jointly. Second, a quality wised algorithm is delivered to develop a hybrid measurement scheme. FPP is applied to obtain the main 3D information. Then, LSS helps to reconstruct the missing depth guided by the quality map. After this, the data fusion algorithm is used to merge and output complete measurement results. Finally, our experiments show significant improvement in the accuracy and robustness of measuring a surface with large reflection variations. In the experimental instance, the accuracy of the proposed method is improved by 0.0278 mm and the integrity is improved by 83.55%.

## 1. Introduction

Three-dimensional (3D) shape information can be widely used in human–computer interaction [1,2], biometric identification [3,4], robot vision [5,6], virtual/augmented reality [7,8], industry [9] and other fields. As a result, 3D shape measurement attracts a lot of attention in the community of computer science and instrument science.

Fringe projection profilometry (FPP) is considered one of the most popular approaches because of the advantages of non-contact operation, high accuracy and full-field acquisition [10,11]. In FPP, sinusoidal fringes are projected onto a measuring surface by using a digital projector. Meanwhile, the observation pattern images are obtained from another angle using a camera. We can decode the height of the surface by analyzing the distortion of the observation fringe patterns [9,12,13]. However, FPP assumes that the measuring surface exhibits a diffuse reflection and usually considers low-reflective (dark) and highlighted (specular reflection) areas as outliers. These regions can completely block any fringe patterns, which results in the loss of depth information [14,15,16]. To address this problem, some solutions are presented. In [17], Salahieh et al. propose a multi-polarization fringe projection (MPFP) imaging technique that handles high dynamic range (HDR) objects by selecting the proper polarized channel measurements. A similar polarization solution is also adopted in [18]. On the other hand, Liu et al. demonstrate the use of a dual-camera FPP system, which can also be considered as two camera-projector monocular systems. By viewing from different angles, these highly specular and dark pixels, which are missing from binocular reconstruction, can be filled in [19]. In addition, Jiang et al. [20] present using 180-degree phase-shifted (or inverted) fringe patterns to reduce the measurement error for high-contrast surfaces reconstruction. Some other researchers have attempted to adjust the parameters of the camera and projector to handle the surface with large reflection variations. Lin et al. [21,22] suggest adjusting the maximum input gray level of projecting image globally, while Chen et al. [23,24] proposed adjusting projecting images pixel-by-pixel. In [16,25], the author proposes projecting a set of fringe images that are captured with different exposures. For [16,21,22,23,24,25], the reflection of the surface needs to be calibrated first. Then, by fusing these images captured with different parameters, a new fringe pattern with fewer saturated regions can be obtained. Although Jiang et al. [16,20,21,22,23,24,25] improve the performance without adding any extra equipment, they need to project or capture a lot of images when the measuring surface has very complex reflection variations. On the other hand, these approaches still use an FPP principle, which can be ineffective for extreme reflection areas.

Laser stripe scanning (LSS) [26,27] is another kind of structured light approach, which shares the same triangulation [28] measurement principle with FPP. The difference is that LSS applies a scanning stripe pattern instead of a fringe pattern. As LSS just needs to extract the stripe in the observation images, it results in very high robustness [29,30,31]. Therefore, LSS is widely used in the 3D shape measurement industry [32]. However, it is expensive to obtain a very high accuracy in LSS, which is mainly determined by the width of the stripe pattern and the resolution of the camera. It is reasonable to consider whether we could use the same hardware to set up an LSS and an FPP system to handle different surface reflection. Generally, because of the limited projection depth of the field, the answer is no. In FPP, Digital Light Processing (DLP) or a Liquid Crystal On Silicon (LCOS) projector is used to project fringe patterns on the measuring surface [33,34]. Those projection techniques can only produce sharp images in the focal plane. If a stripe is projected, it will be severely blurred on the defocused plane. This means that a commercial projector cannot work for LSS. In LSS, a laser stripe projector is adopted, and the object surface is scanned by moving the object or measurement system. A typical laser stripe projector employs a laser beam as the light source, a cylindrical lens is used to scatter the laser beam into a stripe. Therefore, it can’t produce a fringe pattern generally. However, by using Microelectromechanical Systems (MEMS) projection technology, it is possible to generate a stripe pattern and fringe pattern with the same hardware. In MEMS projection [35], a biaxial (or single axial) MEMS micromirror [36,37,38] is applied to scan a laser beam point by point (row by row for single axial MEMS scanner with a cylindrical lens) to produce the projection pattern. In Ref. [39], the authors set up a compact 3D shape measurement system with a single axial MEMS micromirror. As only the FPP principle is used in this work, they still cannot measure the surface with large reflection variations.

In this paper, we propose a hybrid 3D shape measurement approach, which employs FPP and LSS in the same system by applying a biaxial MEMS scanning micromirror to generate the fringe pattern and scanning laser stripe with the same setup. By doing so, the proposed method can handle the surface that has large reflection variations with high accuracy and high robustness.

## 2. Principles

### 2.1. Principle of 3D Shape Measurement with FPP and LSS

#### 2.1.1. Principle of Fringe Projection Profilometry

As shown in Figure 1a, a typical FPP measurement system consists of a digital projector and a digital camera [10]. The light axis (EpO) of the projector intersects the light axis (EcO) of the Charge-coupled Device (CCD) camera at Point O in the reference plane (along the *x*-axis). The distance between the two optical centers is *d*, and the distance between the camera and the reference plane is *l*. Point D is an arbitrary point on the object’s surface with a height of *h*. Points A and C are the intersections of the light paths of the projector and the camera, respectively, with the reference axis. Compared with projecting a sinusoidal fringe pattern onto the reference plane, when an object is placed on the reference plane, the fringe pattern captured by the CCD will be distorted by the object’s height. The modulated phase difference will have a relationship with the true height *h*, as given by Equation (Equation 1), where the inference process can be found in [12]:
(1)$$h_{FPP}\left(x,y\right)=\frac{l\Delta \phi (x,y)}{2\pi f_{0}d},$$
where $f_{0}$ is the frequency of the fringe pattern projection. Δϕ represents the phase difference between Point D and Point A, which is equal to the phase difference between Point A and Point C in the reference plane.

#### 2.1.2. Principle of Laser Stripe Scanning

Laser stripe scanning is based on active laser-triangulation (Figure 1b). In LSS, a laser stripe, created by a dot laser and then scattered by a cylinder lens, is projected onto the measuring object surface and the reflection light is observed under the triangulation angle with a digital camera [26]. Changing the distance between the object and measurement system results in a movement of laser stripe’s position in the *x*-direction observed with the *z*-direction. This position is calculated by extracting the laser stripe center. LSS thus delivers a height distribution of the object. In most cases, industry applications need to make full-field measurements where a highly accurate moving part is introduced. The moving part changes the position between the system and the object, so that the laser stripe sweeps across the surface of the object to obtain full field height distribution. Similar to FPP, when an object is placed on the reference plane, the laser stripe delivers Δx movement, and Equation (Equation 2) shows the relationship between Δx and height $h_{LSS}\left(x,y\right)$:
(2)$$h_{LSS}\left(x,y\right)=\frac{d\Delta x}{l},$$
where *d* is the distance between the laser and the camera. *l* is measurement distance.

### 2.2. Hybrid 3D Shape Measurement System

#### 2.2.1. Biaxial MEMS Micromirror-Based Pattern Projection

The biaxial MEMS micromirror-based pattern projection system is the foundation of our pipeline, which can produce both a stripe pattern and fringe pattern. Figure 2 shows the basic layout of our MEMS pattern projection system. There is a single model laser diode (LD) which served as the light source. Meanwhile, near the LD, an aspheric lens is placed to adjust the focus of the laser beam and shape the beam. Before the laser beam is relayed onto the biaxial MEMS micromirror, there is an aperture to remove the stray light. A biaxial electromagnetic actuation MEMS micromirror working on raster scan mode reflects the light source to the object surface to produce a different 2D pattern. Both of the fast and slow axes rotate reciprocally driven by the current signal, which makes the micromirror scan the laser beam row by row. At the same time, the intensity of the laser is modulated under the synchronization of the sync signal. Figure 3a,b illustrate the controlling signals for fringe pattern and stripe pattern projection. *H* and *V* are the horizontal and vertical driven signals of the MEMS micromirror, respectively. Sync is the row sync signal. Additionally, Laser modulates the LD to produce different intensity. It should be noted that both the horizontal and vertical axes just operate in resonant vibration mode, no matter whether the fringe pattern or stripe pattern is projected. As shown in Figure 3, *H* and *V* are always sinusoidal waveforms. This kind of variable-speed scanning introduces distortion in pattern projection, which can be considered as a phase error resulting in a systematic error. To remove the distortion, pre-correction is performed on Laser generally, where more detailed information can be found in [35].

Different from the pixel array based projection technique, a MEMS micromirror based pattern projection system produces different patterns by scanning the laser beam two-dimensionally. This makes it possible to project fringe pattern and stripe pattern with the same hardware. Meanwhile, due to a laser source having better linearity than the light-emitting diode (LED) source, no gamma correction [40] is required in FPP with proposed pipeline, which brings additional benefits.

#### 2.2.2. Quality Index in the Proposed Approach

Our system is significantly simplified by implementing FPP and LSS in the same system. Another benefit is that the data from FPP and LSS are based on the same coordinate system, which makes it much easier to align these two parts of the data. Before data fusion, we need to build an error model to evaluate the quality of measuring data, so that we can guide the process of data fusion. In this section, we define a quality index for our hybrid measurement approach.

In FPP, the phase of each point is calculated by the phase shift method. The fringe image obtained by the CCD can be described by
(3)Ici(x,y)=a0(x,y)+bmod(x,y)cos(Φ+2πni2πniNsNs)+n(x,y).

In Equation (Equation 3), $a_{0}\left(x,y\right)$ is the respective backgrounds, while $b_{mod}\left(x,y\right)$ is the respective modulation functions, also called the contrast. In addition, $N_{s}$ is the number of steps of phase shifting, $n_{i}$ is an integer, and n(x,y) is the random noise.

In fact, we modulate the phase by changing the grayscale of the projection images. Here, we would like to discuss how the grayscale affects measurement accuracy. In the N-steps phase shifting method, the phase shifting noise caused by random noise is determined by $N_{s}$ and the distribution of n(x,y). We assume that this part noise $n_{s}\in \left[-N,N\right]$, as shown in Figure 4, where the complex plane represents the phase calculated by an imaginary part and a real part. If point P is a measurement point, the coordinates of Point P are $P(b_{mod}\cos\Phi ,b_{mod}\sin\Phi )$. Due to the noise $n_{s}$, P will and change within the blue square (Figure 4), which has a side length of 2N. If n=(−N,N), then P will shift to Q and P will have a maximum phase error of $\varphi_{\max}$. If we assume that the phase of P is Φ, then the phase of Q can be given by
(4)$$\Phi +\varphi =\arctan(\frac{b_{mod}\sin\left(\Phi \right)+N}{b_{mod}\cos\left(\Phi \right)-N}).$$

Thus, the phase difference is φ. Supposing that $k=\frac{N}{b_{mod}}\in \left[0,1\right)$, then Equation (Equation 4) can be simplified as
(5)$$\varphi =\arctan\left(\frac{\sin(\Phi )+k}{\cos(\Phi )-k}\right)-\Phi .$$

From Equation (Equation 5), we know that the error of the FPP system is determined by *k*. This means that we can evaluate FPP depth data by $b_{mod}$.

In a fixed experiments setup, the noise in Equation (Equation 3) considered as a constant variable generally. Therefore, it can be simplified as Equation (Equation 6):
(6)$$\begin{array}{cc} I_{ci}\left(x,y\right) & =A+b_{mod}\cos\left(\Phi +2\pi n_{i}/N_{s}\right)\\ & =A+b_{mod}\cos\left(\Phi +\delta_{i}\right)\\ & =A+b_{mod}\left(\cos\Phi \cos\delta_{i}-\sin\phi \sin\delta_{i}\right)\\ & =A+B_{c}\cos\delta_{i}+B_{s}\sin\delta_{i}, \end{array}$$
where *A* is the combine of background $a_{0}\left(x,y\right)$, noise n(x,y), and $b_{mod}$ is the modulation in FPP:
(7)$$\left\{\begin{array}{c} A=a_{0}\left(x,y\right)+n\left(x,y\right),\\ B_{c}=b_{mod}\cos\phi ,\\ B_{s}=-b_{mod}\sin\phi ,\\ \delta_{i}=i\frac{2\pi }{N_{s}};i=0,1\cdots N_{s}-1. \end{array}\right.$$

Based on the principle of trigonometric function orthogonality,
(8)$$\left\{\begin{array}{c} B_{c}=\frac{2}{N_{s}}\sum I_{i}\cos\delta_{i},\\ B_{s}=\frac{2}{N_{s}}\sum I_{i}\sin\delta_{i}. \end{array}\right.$$

Thus, we have
(9)$$b_{mod}=\sqrt{{B_{c}}^{2}+{B_{s}}^{2}}.$$

It can be known from Equation (Equation 5) that the quality of depth data is positively correlated with the modulation of fringe images. Therefore, we choose to conclude that modulation as the guiding quality index for data fusion in the proposed pipeline, where the quality index *Q* can be defined in Equation (Equation 10):
(10)$$Q=\sqrt{{\left(\frac{2}{N_{s}}\sum I_{ci}\cos\delta_{i}\right)}^{2}+{\left(\frac{2}{N_{s}}\sum I_{ci}\sin\delta_{i}\right)}^{2}}.$$

The region with high reliability, where the *Q* is above a threshold, will use the depth data from FPP. As a supplement, depth information that comes from LSS is used to fill in the other part, where the *Q* is not good enough. This allows us to keep the integrity with the optimal accuracy.

#### 2.2.3. Hybrid 3D Shape Measurement Pipeline

Our Hybrid 3D shape measurement pipeline is shown in Figure 5. Both FPP and LSS are implemented with one system. Before measuring, we should first calibrate the camera and the projector. The calibration process provides the relationship between the height and distortion of structured Pattern [41]. FPP is adopted to obtain the quality map and depth map 1, while LSS is employed to obtain depth map 2. Generally, depth 1 has high accuracy but loses some depth information because of the extreme reflection. On the other hand, map 2 has lower accuracy but loses little depth information due to excellent robustness. Because map 1 and map 2 are naturally aligned, we can fill the final depth map by selecting a better part from map 1 and map 2 without any registration. The quality map shows where FPP works well; therefore, it is employed to guide the data fusion. When the quality index in quality map is above a threshold, the depth coming from FPP is considered high quality and will be used for the final depth map. Otherwise, LSS data will be adopted, and even both FPP and LSS data are available. Finally, the 3D surface is reconstructed from the refined depth map and calibration parameters. A carefully selected threshold is very important for proposed pipeline, and it will be presented in the next section.

## 3. Experiments

To implement the proposed hybrid measurement approach, we build our system as in Figure 2. Figure 6 illustrates our experimental setup. A MEMS pattern projector, driven by the driver board, produces both the fringe pattern and stripe pattern. In addition, a USB hub is adopted for communication and collection of the image data. A CCD camera with a 12 mm lens is used to capture the pattern images. The angle between the MEMS projector and the camera is set to 15 degrees, which can balance measuring resolution and coincidence of field of view. While the measuring distance is set to 500 mm, where the projector and CCD have the largest coincidence field of view and the best projection pattern quality. All these modules are mounted on an aluminum alloy casing with the overall size of 187 mm × 90 mm × 45 mm. More detailed parameters and description for these components are in Table 1. Both camera, USB hub and lenses are commercial products. The optical and mechanical part are designed by our own team. The driver board and MEMS scanner are designed and manufactured by the cooperation team. With all these setups, the hybrid 3D shape measurement system achieves a 0.07 mm measuring accuracy with a 286 mm × 176 mm field of view at an optimum working distance of 500 mm. Additionally, laser beam scanning has a larger depth of field than pixel array based projection. Hence, the depth of view of proposed system is mainly determined by the limitation of camera. It is about 100 mm in this experiment.

### 3.1. Linearity Test of Proposed Pattern Projection System

In FPP, the nolinearity, which makes ideal sinusoidal waveforms nonsinusoidal, can significantly influence the performance [40]. Due to the use of a laser source, the proposed system has better linearity than a conventional LED based pattern projector. To verify the linearity of these two kinds of projectors, a commercial projector (coolux S3) is chosen as a comparison, which is an LED based DLP projector. Both projectors project pure white images with different gray levels. Then, an illuminometer is used to measure the illuminance with a fixed distance. Figure 7a,b illustrate the relationship between the projection gray value and illuminance of the proposed system and a commercial LED projector. As is shown in Figure 7, the proposed system has better linearity than a commercial LED projector. In fact, these two curves give luminescence characteristics of laser source and LED source. Therefore, gamma calibration, which is usually applied to eliminate the influence of nonlinearity in FPP [40], is no longer needed.

### 3.2. Experiments on the Quality Index

In the previous section, we demonstrated the use of the quality index to evaluate depth information, and give the theoretical basis. In this section, we will verify the given theoretical basis and find the best threshold for data fusion by experiments. Here, we use a standard gauge block (30 mm × 50 mm × 8 mm) as the measuring object. The flatness of the block is less than 5 µm. The distribution of the depth value reflects the quality of depth data. In this work, we take the uncertainty of depth data as the evaluation of the measurement accuracy. Additionally, it is computed with Equation (Equation 11),
(11)$$RMS_{error}=\sqrt{\frac{1}{N}\sum_{i=1}^{N}{\left(e_{i}-\bar{e}\right)}^{2}},$$
where $e_{i}$ is the height of real measuring points and $\bar{e}$ is the height of the same position in the fitted plane by Equation (Equation 12) with the least squares method. *N* is the total number of real measuring points, and *i* stands for the index of a point:
(12)$$A_{fit}x+B_{fit}y+C_{fit}z+D_{fit}=0.$$

To find the relationship between fringe modulation and measurement accuracy, we set the modulation of the projected fringe pattern as different values to obtain the related uncertainty. Figure 8a is the standard gauge block. Figure 8b is the standard gauge block with the fringe pattern. Figure 8c illustrates the relationship between the modulation of the captured fringe pattern and measurement error. Obviously, lower modulation results in higher measurement error.

In Equation (Equation 2), the uncertainty of $h_{LSS}\left(x,y\right)$ is determined by the resolution of Δx due to the Δx being observed with a CCD camera. Thus, the resolution of Δx is a spatial distance, which is one pixel in the observation frame. It can be calculated as
(13)$$\Delta x_{\min}=\frac{l\delta }{f},$$
where *l* is the measurement distance, δ and *f* are the size of pixel and the focal length of the lens of observation camera. In these experimental settings, l=500 mm, δ=0.0064 mm and f=12 mm. Jointly with Equation (Equation 2), we have $\Delta h_{LSS}\left(x,y\right)$ = 0.0715 mm. From Figure 8, when the modulation is above 32, FPP presents better accuracy than 0.0715 mm, which is the resolution of LSS. Therefore, the threshold for data fusion is chosen to be 32.

### 3.3. Hybrid 3D Shape Measurement

In this section, a black porcelain doll is chosen as our measuring object—see Figure 9a. For a black porcelain surface, the normal direction has a strong influence on the reflection. Weak reflection can be observed in most regions. When the normal direction is towards the camera, extremely strong reflection will be captured. This makes it difficult for FPP to work well. As a comparison, we paint the body of the doll with developer, which makes the body an ideal diffuse surface. Figure 9b is the quality map, where the threshold is 25. Figure 9c illustrates the fusion mask. Guided by the mask, we fuse the depth of FPP Figure 10a with the depth of LSS Figure 10b, and then we obtain the optimal 3D shape information Figure 10c. From Figure 10, we can find that FPP gives better accuracy (body of doll), but it cannot handle large reflectance variations (head of doll). LSS shows high robustness with different reflectance (head of doll) but lower accuracy (body of doll). The fusion data combines the advantages of FPP and LSS. To evaluate the measurement data quantitatively, a 3D measurement instrument (see Figure 11a) with a resolution of ±5 µm, is adopted to offer the ground truth data (see Figure 11b). Because of the limitation of measurement range and efficiency, only the head of the doll is scanned. Then, the FPP reconstruction result and hybrid measurement result are registered with ground truth data to compute the 3D geometric error, respectively. As illustrated in Figure 12a, for FPP data, the root mean square (RMS) error is 0.0991 mm, and 0.0713 mm for hybrid measuring results in Figure 12b. In this case, integrity is computed to evaluate the robustness. For FPP, it is 54.48% taking hybrid results as the reference (100%). The experiments show that the proposed hybrid approach gets an improvement of 0.0278 mm in accuracy and 83.55% in integrity under the conditions described in this paper.

To verify the performance of the proposed approach, we chose several objects with large reflection variations that FPP cannot handle integrally. Figure 13a shows a plastic car model with shiny, dark and specular reflection regions. Figure 13b shows a metal surface. When the normal direction changes dramatically, it becomes hard to scan a metal surface. Figure 13c,d also show two difficult cases: stone material with very low reflection and a plastic surface with multiple reflectance and a large variation of normal directions. These results show the excellent performance of our approach.

## 4. Conclusions

In this paper, we have addressed the 3D shape measurement for large reflection variations with a hybrid approach. We proposed using a biaxial MEMS scanning micromirror and laser source to produce the fringe pattern and stripe pattern with the same hardware. Both FPP, which has the advantages of high accuracy and high efficiency, and LSS, which is one of the most robust methods, are employed to achieve a hybrid 3D shape measurement approach. Real experiments of different objects with large reflectance variations were carried out to verify the proposed method. The metal, plastic and stone materials with large reflection variations and large normal direction variations were reconstructed successfully, which shows the excellent performance of our method.

## Figures and Tables

**Figure 1 micromachines-10-00047-f001:**
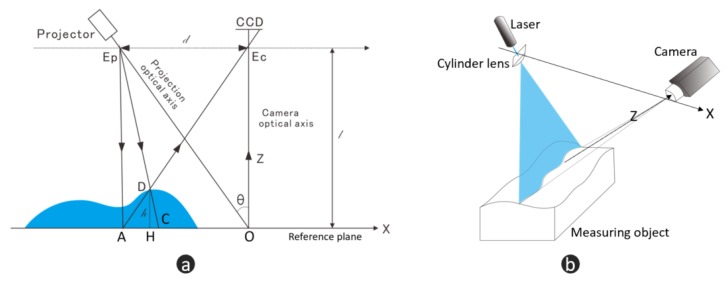
Schematic drawings of fringe projection and laser stripe scanning. (**a**) basic light path of Fringe projection profilometry system; (**b**) schematic diagram of laser stripe scanning system.

**Figure 2 micromachines-10-00047-f002:**
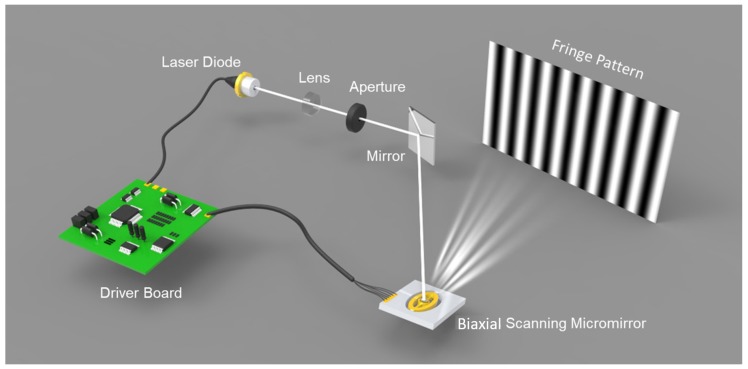
Schematic diagram of the biaxial Microelectromechanical Systems (MEMS)-based fringe pattern projection system.

**Figure 3 micromachines-10-00047-f003:**
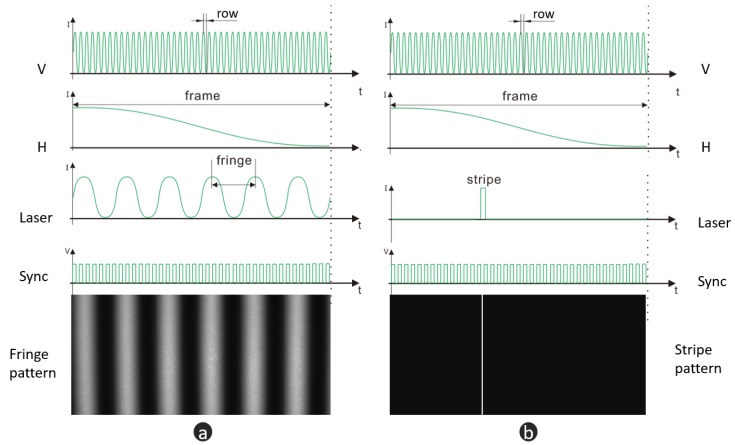
Controlling signals of pattern projection. (**a**) controlling signal of fringe projection; (**b**) controlling signal of stripe projection. *H* and *V* are the horizontal and vertical driven signals of the MEMS micromirror. Sync is the row sync signal. Laser is the modulation signal of the laser diode (LD).

**Figure 4 micromachines-10-00047-f004:**
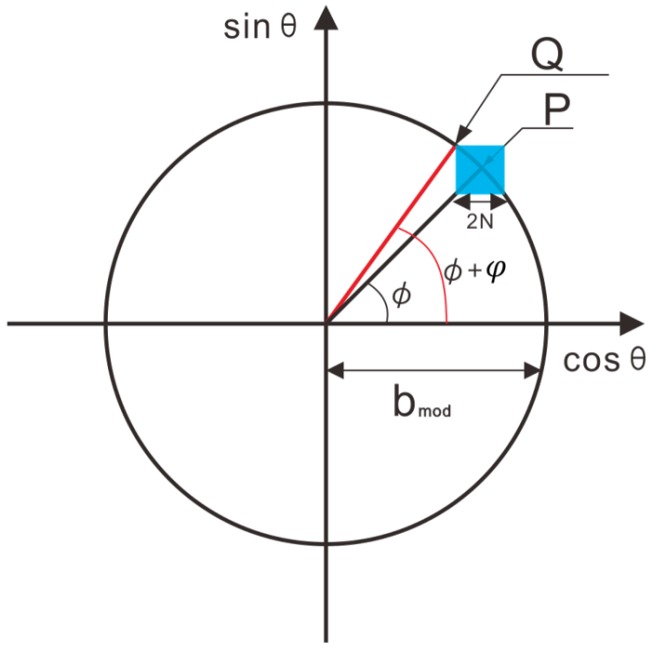
Error analysis in fringe projection profilometry (FPP).

**Figure 5 micromachines-10-00047-f005:**
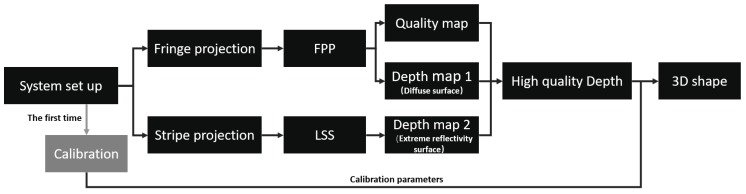
Flow chart of the proposed hybrid measurement approach.

**Figure 6 micromachines-10-00047-f006:**
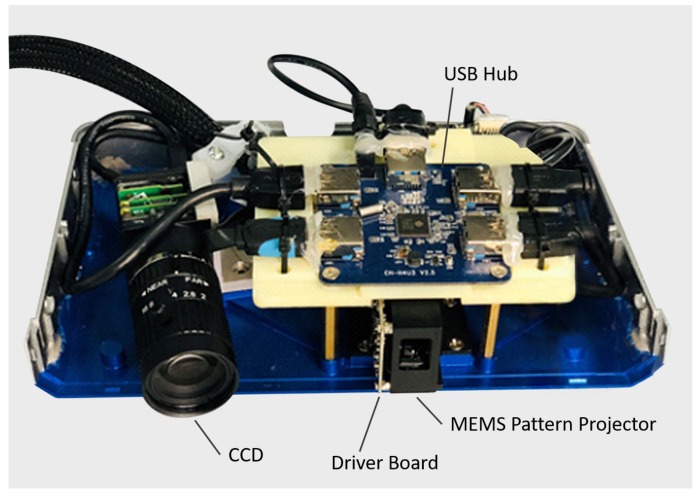
Hybrid 3D shape measurement setup.

**Figure 7 micromachines-10-00047-f007:**
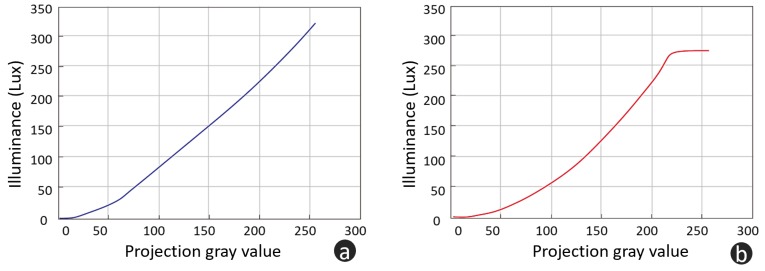
Linearity test. (**a**) proposed system; (**b**) commercial LED projector, item model: coolux S3.

**Figure 8 micromachines-10-00047-f008:**
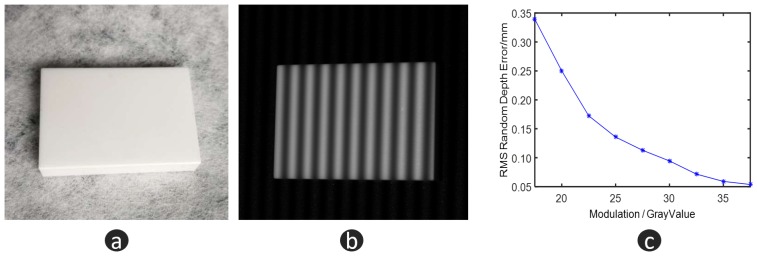
Depth reliability experiments. (**a**) standard gauge block; (**b**) standard gauge block with fringe pattern; and (**c**) the relationship between modulation and measurement error.

**Figure 9 micromachines-10-00047-f009:**
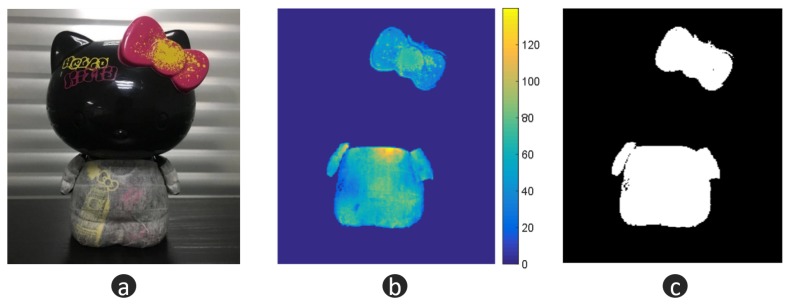
Measuring object and intermediate data. (**a**) measuring object; (**b**) quality map from FPP; (**c**) binary mask for data fusion.

**Figure 10 micromachines-10-00047-f010:**
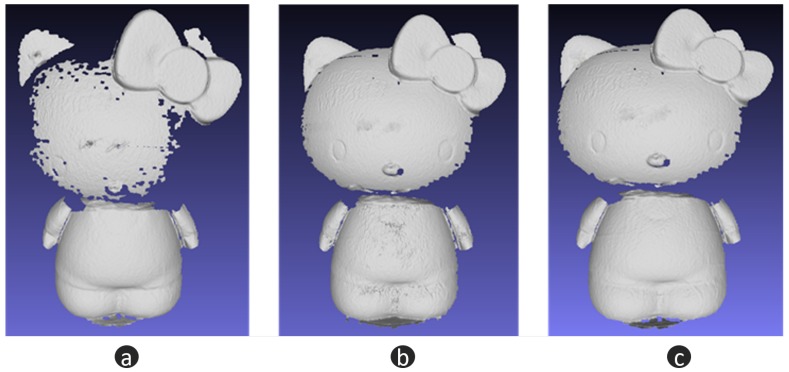
3D reconstruction results. (**a**) reconstructing from FPP; (**b**) reconstructing from laser stripe scanning (LSS); (**c**) fusion results.

**Figure 11 micromachines-10-00047-f011:**
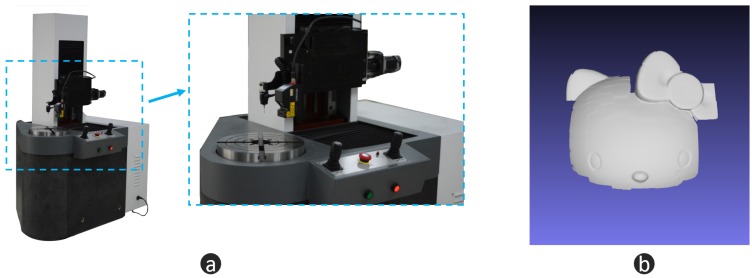
Ground truth data. (**a**) high accuracy 3D measurement equipment, resolution = ±5 µm; (**b**) high accuracy ground truth measurement result.

**Figure 12 micromachines-10-00047-f012:**
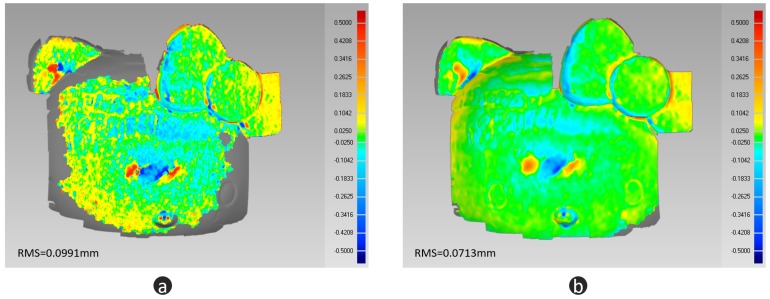
Accuracy evaluation. (**a**) comparison between FPP data and ground truth; (**b**) comparison between the data measured with proposed method and ground truth.

**Figure 13 micromachines-10-00047-f013:**
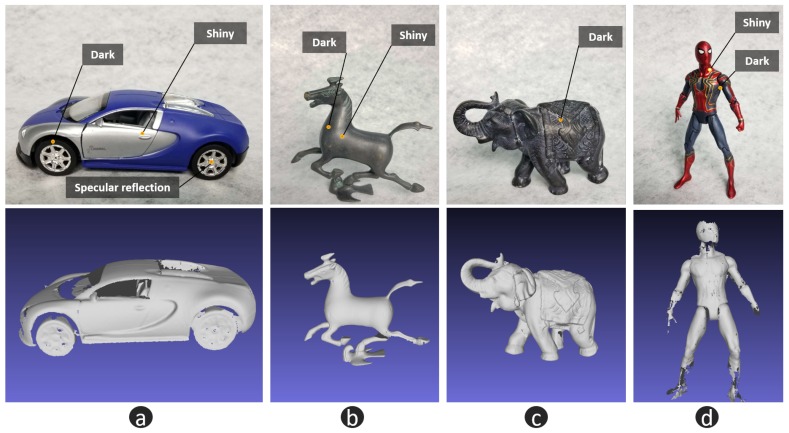
Experimental results of the proposed hybrid measurement approach. (**a**) plastic surface with shiny, dark and specular reflection regions; (**b**) metal surface, which results in shiny and dark regions from different perspectives of observation; (**c**) stone material with very low reflection; (**d**) plastic surface with multiple reflectance and large variation of normal directions.

**Table 1 micromachines-10-00047-t001:** Parameters and description for experimental components.

Items	Parameters and Description
Camera	Model: Charge-coupled Device (CCD), mono, global shutter. Resolution: 1920 × 1200. Max frames per second: 163
Lens for camera	Focal length: 12 mm
MEMS scanner	Model: 2D electromagnetic actuation. Fast axis: 18 kHz, ±16°. Slow axis: 0.5 kHz, ±10°
Laser diode	Wavelength: 650 nm. Power consumption: 320 mW
Lens for LD	Model: aspheric lens. Focal length: 4.51 mm. Aspheric coefficient: −0.925. Distance from LD: 4.55 mm. Distance from MEMS scanner: 33 mm
USB hub	USB3.0 × 5

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
