# Peer review of "Hybrid 3D Shape Measurement Using the MEMS Scanning Micromirror"

_micromachines, 2019, doi:10.3390/mi10010047_

Round 1
Reviewer 1 Report
Authors present a hybrid approach to measure Surface shape. They merge LSS and FPP systems into a single system operating in two regimes simultaneously. The work is interesting however it need further clarification.
1. Please check English for correctness, e.g., words ‘row’ and ‘raw’ are used incorrectly.
2. Please address the issue of fringe pattern nonlinearity.
3. Please discuss in depth the modulation threshold selection for the modulation guided merging procedure of FPP and LSS data.
4. Please discuss in depth differences in FPP and LSS data in areas when both data are available.
5. Define the resolution of the hybrid system and other important parameters like depth of field, field of view etc.
Author Response
We sincerely thank the reviewer for constructive criticisms and valuable comments, which
were of great help in revising the manuscript. Accordingly, the revised manuscript has been
systematically improved with new information and additional interpretations. Our responses
are given in the attachment.

Reviewer 2 Report
Please see attached review file

Author Response

(The authors gave the same response as above.)

Round 2
Reviewer 1 Report
Thank you for addressing my remarks in a satisfactory manner. I recommend publication.
Reviewer 2 Report
Thank you for adapting the manuscript based on the reviewers comments. The added details and explanations lead to a significant improvement and I am happy to recommend the manuscript for publication once a couple of minor things have been addressed:
- The added background in the introduction leads to a non-sequential numbering of references. Please adjust the reference numbering to follow common procedure of reference numbers following a sequential order upon first mention
- In Figure 3, the label on graph H should be "row" not "raw". Additionally, with H usually being horizontal and V being vertical, I would expect the fringe pattern to be 90deg rotated (or the label of H and V be swapped) for both the fringe projection and stripe projection.